# A SIMPLE BUT EFFECTIVE AND EFFICIENT GLOBAL MODELING PARADIGM FOR IMAGE RESTORATION

## ABSTRACT

Global modelling-based image restoration frameworks (*e.g.*, transformer-like architecture) have gained popularity. Despite the remarkable advancement, their success may be at the cost of model parameters and FLOPs while the intrinsic characteristics (*e.g.*, the task-specific degradation) are ignored. The objective of our work is orthogonal to previous studies and tailors a simple yet effective and efficient global modelling paradigm for image restoration. The key insights which motivate our study are two-fold: 1) Fourier transform is capable of disentangling image degradation and content component, serving as the image degradation prior embedded into image restoration framework; 2) Fourier domain innately embraces global property where each pixel of Fourier space is involved with all spatial pixels. We obey the de facto global modeling rule "spatial interaction + channel evolution" of previous studies. Differently, we customize the core designs: Fourier spatial interaction modeling and Fourier channel evolution. Equipped with the above-mentioned designs, our image restoration paradigm is verified on mainstream image restoration tasks including image de-raining, image enhancement, image de-hazing, and guided image super-resolution. Extensive experiments suggest that our paradigm achieves the competitive performance with fewer computational resources. Our main focus is not to beat previous frameworks but provide an alternative global modeling-based customized image restoration framework with efficient structure. Code will be publicly available.

## 1 INTRODUCTION

Image restoration aims to recover the latent clear image from its given degraded version. It is a highly ill-posed and challenging issue as there exists infinite feasible results for single degraded image. The representative image restoration tasks include image de-raining, image de-hazing, low-light enhancement, guided image super-resolution, etc.

In the past decades, a mount of research efforts have been devoted to solving the single image restoration problem, which can be classified into two categories: traditional optimization methods and deep learning-based methods (Zhang et al., 2018; Ren et al., 2018; Zhang et al., 2018; Ren et al., 2016b; Fu et al., 2021; Zhang et al., 2020; Liu et al., 2021a). In terms of traditional image restoration methods, they formulate the image restoration process as an optimization problem and develop various image priors of the expected latent clear image to constrain the solution space, *e.g.*, dark channel prior for image de-hazing (Dark, 2009), histogram distribution prior for underwater image enhancement (Li et al., 2016), non-local mean prior for image de-noising (Dixit & Phadke, 2013), sparse image prior for guided image super-resolution (Kim & Kwon, 2010) as well as the commonly-used local and non-local smooth prior (Chen et al., 2013), low-rank prior (Ren et al., 2016a). However, aforementioned image priors are difficult to develop and these traditional methods involve the iteration optimization, thus consuming the huge computational resources and further hindering their usage. In a word, the common sense is to explore the potential image prior to relieve the optimization difficulty of the ill-posed image restoration.

On the line of deep learning-based methods, convolutional neural networks (CNNs) have received widespread attention and achieved promising improvement in image restoration tasks over traditional methods (Liu et al., 2020; Ma et al., 2021; Zhang et al., 2021a; Zhou et al., 2021; 2022b). More recently, transformer and multi-layer perceptrons (MLPs)-based global modeling paradigms

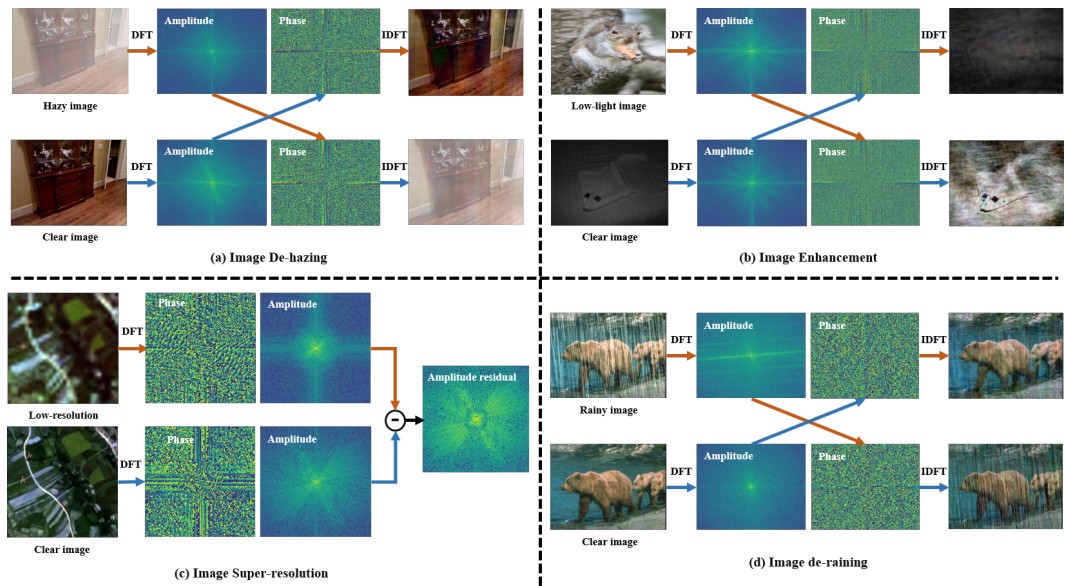

Figure 1: Motivations. Analysis of discrete Fourier transform (DFT) over mainstream image restoration tasks. In (a) and (d), we respectively swap the amplitude component and phase component of a degraded image and its clear version. It can be observed that the degradation effect is transferred, thus indicating that Fourier transform is capable of disentangling image degradation and content component and the degradation mainly lies in the amplitude component. To further verify our observation, we also swap the amplitude component and phase component of a degraded image and an irrelevant image in (b). The degradation is still mainly related to the amplitude component, such as the darkness for image enhancement. Similarly, a low-resolution image and its high-resolution counterpart are different in the amplitude component in (c). These observation motivates us to leverage the Fourier transform as the image degradation prior embedded into image restoration framework. More analysis and results can be found in the Appendix.

have struck the image restoration field and significantly surpassed the CNN-based methods. Despite the remarkable advancement, they are arbitrarily used for image restoration tasks while ignoring the intrinsic characteristics of specific image restoration task. The success may be owing to the huge cost of computational resources, limiting their practical applications, especially on resource-limited devices. We therefore wonder "Can we provide a customized global modeling image restoration paradigm in a simple but effective and efficient manner?"

To this end, motivated by our observations on Fourier transformation for image restoration tasks in Figure 1, we tailor a simple yet effective and efficient global modelling paradigm, which is orthogonal to previous studies and customized for image restoration. The core insights of our work are two-folder: 1) **general image restoration prior**: Fourier transform is capable of disentangling image degradation and content component, serving as the image degradation prior embedded into image restoration framework; 2) **global modeling**: Fourier domain innately embraces global property where each pixel of Fourier space is involved with all spatial pixels. As shown in Figure 2, the existing global modeling paradigm (*e.g.*, transformer and MLP-Mixer) follow the the de-facto global modeling rule "spatial interaction + channel evolution". Similarly, we obey the rule and customize the core designs: **Fourier spatial interaction and Fourier channel evolution**. Such designs are different from previous works and provide new insights on global modeling network structures for image restoration. Equipped with the above-mentioned designs, our image restoration paradigm tailed for image restoration is described in Figure 3. Extensive experiments are conducted on mainstream image restoration tasks including image de-raining, image enhancement, image de-hazing, and guided image super-resolution. Experimental results suggest that our paradigm achieves the competitive performance with fewer computational resources. To emphasize, *our main focus is not to beat previous frameworks but provide an alternative global modelling-based customized image restoration framework with efficient structure.*

Our contributions are summarized as follows: **(1)** We contribute the first global modeling paradigm for image restoration in a simple but effective and efficient manner. **(2)** We implicitly embed the

Fourier-based general image degradation prior into our core structures: Fourier spatial modeling and Fourier channel evolution, which provides new insights on the designs of global modeling-based image restoration network. **(3)** Our proposed paradigm achieves the competitive performance on several mainstream image restoration tasks with fewer computational resources.

## 2 RELATED WORK

**Image restoration.** Image restoration aims to restore an image degraded by degradation factors (e.g. rain, haze, noise, lowlight) to a clear counterpart, which has been studied for a long time. Traditional image restoration methods are usually designed as an optimization problem, which incorporate specific priors of latent clear image to constrain the solution space (Dark, 2009; Li et al., 2016; Dixit & Phadke, 2013; Kim & Kwon, 2010). For example, dark channel prior (Dark, 2009) is proposed for image dehazing and histogram distribution prior (Li et al., 2016) is developed for underwater image enhancement. These methods involve iteration optimization, thus consuming the huge computational resources and limiting their application. Recently, deep learning-based methods have achieved impressive performance in a data-driven manner. Among them, most algorithms are designed with CNN-based architectures. Early works stack deep convolution layers for improv-

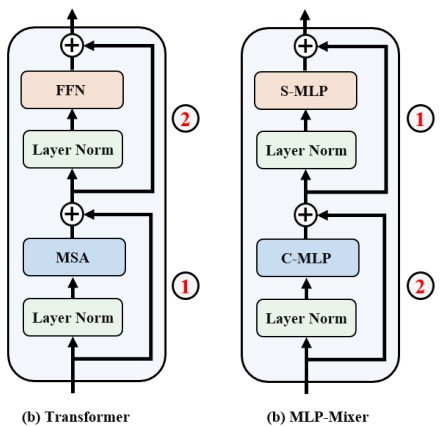

Figure 2: The underlying rule of existing global model paradigm: spatial interaction + channel evolution.

ing model representation ability, such as VDSR (Kim et al., 2016), DnCNN (Zhang et al., 2017), and ARCNN (Dong et al., 2015). Based on them, advanced methods have adopted more powerful architecture designs, such as residual block (Tai et al., 2017; Ehrlich & Davis, 2019) and dense block (Zhang et al., 2020; Dong et al., 2020). Besides, attention mechanisms (Zhang et al., 2018; 2021c) and multi-stage mechanism (Zamir et al., 2021; Chen et al., 2021c) have brought into image restoration algorithms that elevate the performance. However, the locality property of convolution operation limits the perception of global information that is critical for image restoration (Dixit & Phadke, 2013; Berman et al., 2016).

**Global modeling.** In recent years, global modeling techniques have gained much popularity in the computer vision community. A line of these methods is based on transformer (Vaswani et al., 2017), which has been adapted in numerous vision tasks such as vision recognition (Liu et al., 2021b; Xia et al., 2022) and segmentation (Chen et al., 2021b; Cao et al., 2021). Different from CNN-based architectures, transformer learns long-range dependencies between image patch sequences for global-aware modeling (Dosovitskiy et al., 2020). Due to its characteristic, various image restoration algorithms based on transformer have been proposed in recent years, which achieve superior performance in restoration tasks such as image image dehazing (Chun-Le Guo, 2022), image deraining (Xiao et al., 2022) and low-light image enhancement (Xu et al., 2022). Among them, a pioneer work IPT directly applies vanilla transformers to image patches (Chen et al., 2021a), while Uformer (Wang et al., 2022b) and SwinIR (Liang et al., 2021) apply efficient window-based local attention models on several image restoration tasks. However, the huge computation cost and parameters of transformer framework limit practical application. As another line of global modeling paradigm, multi-layer perceptrons (MLPs)-based methods have attracted attention in vision problems (Tolstikhin et al., 2021). To adapt this architecture for image restoration problems, MAXIM adopts a multi-axis MLP based mechanism to perceive information with global receptive field (Tu et al., 2022b). Nevertheless, it still costs enormous computation resources and is thus hard to apply in compact devices. In total, all above architectures are not fully to explore priors that are specific for image restoration tasks, which is important to lift performance. Recently, Fourier transformation has presented its effectiveness for global modeling (Chi et al., 2019; 2020). Instead of further exploring the efficacy of Fourier as global modeling in high-level tasks such as image classification, video action classification, human keypoint detection in (Chi et al., 2019), our work is the first to

focus on the customized image restoration framework designs. The work proposed in (Chi et al., 2019) pays more attention to the global property while our framework further explores the intrinsic prior tailored for image restoration. In addition, different from existing Fourier techniques Chi et al. (2020) that emphasize the micro basic operator with the global receptive field, our work aims to focus on the macro framework design. In our work, we pay more attention to the customized image restoration global modeling framework. In this work, we investigate to incorporate restoration prior with Fourier transformation to conduct effective global modeling, which is efficient for practical application.

Different from existing transformer-based Wang et al. (2022a); Zamir et al. (2022) and MLP-based methods Tu et al. (2022a) that do not contain the intrinsic knowledge about image restoration tasks and only roughly focus on the global operator designs, our proposed framework is the first to explore the customized image restoration global modeling paradigm. Unlike these works that only consider global modeling, our work with efficient structure also meets the requirement of image restoration on edge devices with limited computation sources. In a word, our proposed framework incorporates both advantages of the global modeling mechanism and general image degradation prior that are introduced by Fourier transformation, thus achieving better performance.

## 3  METHOD

In this section, we first revisit the properties of Fourier transformation for image and then present an overview of the proposed global modeling paradigm, as illustrated in Figure 3. We further provide details of the fundamental building block of our method. Finally, we deep into the new loss functions proposed in our work.

### 3.1  PRELIMINARY OF FOURIER TRANSFORMATION FOR IMAGE

As recognized, the Fourier transform is widely used to analyze the frequency content of images. For the images of multiple color channels, the Fourier transform is calculated and performed for each channel separately. For simplicity, we eliminate the notation of channels in formulas. Given an image $x \in R^{H \times W \times C}$, the Fourier transform $\mathcal{F}$ converts it to Fourier space as the complex component $\mathcal{F}(x)$, which is expressed as:

$$\mathcal{F}(x)(u,v) = \frac{1}{\sqrt{HW}} \sum_{h=0}^{H-1} \sum_{w=0}^{W-1} x(h,w) e^{-j2\pi(\frac{h}{H}u + \frac{w}{W}v)}, \tag{1}$$

$\mathcal{F}^{-1}(x)$ defines the inverse Fourier transform accordingly. Both the Fourier transform and its inverse procedure can be efficiently implemented by FFT/IFFT algorithms (Frigo & Johnson, 1998). The amplitude component $\mathcal{A}(x)(u,v)$ and phase component $\mathcal{P}(x)(u,v)$ are expressed as:

$$\mathcal{A}(x)(u,v)) = \sqrt{R^2(x)(u,v)) + I^2(x)(u,v))},$$
$$\mathcal{P}(x)(u,v)) = \arctan[\frac{I(x)(u,v))}{R(x)(u,v))}], \tag{2}$$

where $R(x)$ and $I(x)$ represent the real and imaginary part respectively. Note that the Fourier transformation and inverse procedure are computed independently on each channel of feature maps.

Targeting at image restoration, we employ Fourier transformation to conduct the detailed frequency analysis by revisiting the properties of phase and amplitude components, as shown in Figure 1. It can be observed that the degradation effect is transferred (mainly in the amplitude component) when swapping the amplitude component and phase component of a degraded image and its clear version. The phenomenon indicates that Fourier transform is capable of disentangling image degradation and content component and the degradation mainly lies in the amplitude component. This motivates us to leverage Fourier transform as the image degradation prior embedded into image restoration framework.

### 3.2  FRAMEWORK

**Structure flow**. Our main goal is to develop a simple but effective and efficient global modeling paradigm for image restoration in a U-shaped hierarchical architecture, detailed in Figure 3. Given a

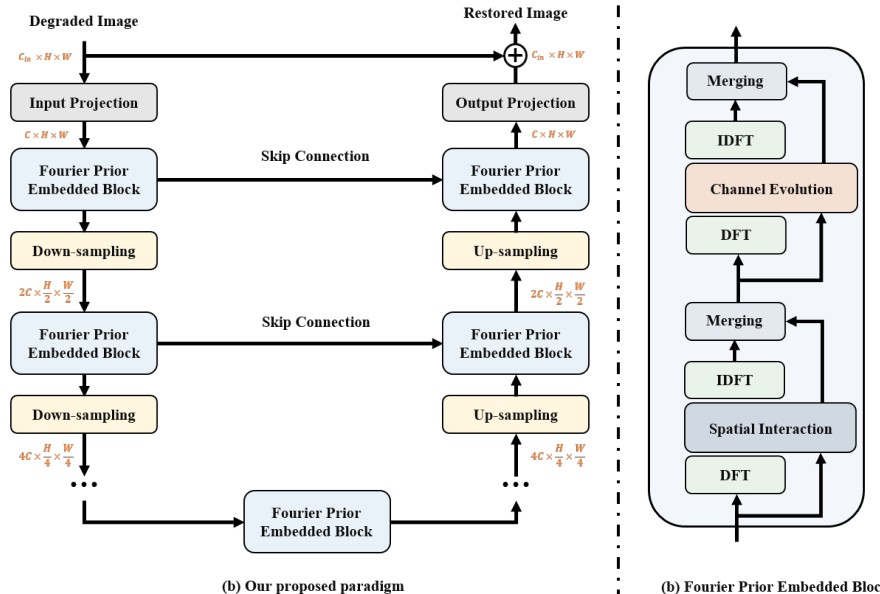

Figure 3: Overview of the proposed customized global modeling paradigm for image restoration.

degraded image $\mathbf{I} \in \mathrm{R}^{\mathrm{H} \times \mathrm{W} \times \mathrm{C_{in}}}$, the proposed paradigm first applies the convolution layer to project $\mathbf{I}$ into shallow feature embedding $\mathbf{X_0} \in \mathrm{R}^{\mathrm{H} \times \mathrm{W} \times \mathrm{C}}$. Next, following the U-shaped network designs, the obtained shallow embedding is passed through $N$ encoder stages where each stage consists of a stack of the proposed core building module dubbed as Fourier Prior embedded Block and one down-sampling layer. The Fourier Prior embedded Block takes advantage of the inborn global modeling property of Fourier transform, and obeys the underlying global modeling rule "spatial interaction + channel evolution" to customize the Fourier spatial and channel information interaction. In the downsampling layer, we downsample the 2D spatial feature maps using 3 × 3 convolution with stride 2. Similarly, in decoder stages, we employ the stack of the proposed Fourier Prior embedded Block and one upsampling layer for feature reconstruction in each stage. To assist the recovery process, each stage takes the high-level decoder features concatenated with the same stage low-level encoder features via skip connections as input. It is beneficial in preserving the fine structural and textural details in the restored images. Finally, a convolution layer is applied to the refined features to generate residual image $\mathbf{I} \in \mathrm{R}^{\mathrm{H} \times \mathrm{W} \times \mathrm{C_{in}}}$ to which degraded image is added to obtain the final restored image $H_O$.

**Optimization flow**. Besides the network designs for image restoration, we also introduce a new loss function to enable the network for better optimization, thus reconstructing the more pleasing results in both spatial and frequency domains. In detail, it consists of two parts: spatial domain loss and frequency domain loss. In contrast to existing methods that usually adopt pixel-level losses with local guidance in the spatial domain, we additionally propose the frequency domain supervision loss via Fourier transformation that is calculated on the global frequency components. Motivated by spectral convolution theorem, direct emphasis on the frequency content is capable of better reconstructing the global information, thus improving the restoration performance.

Let $H_O$ and $GT$ denote the network output and the corresponding ground truth respectively. We propose a joint spatial-frequency domain loss for supervising the network training. In spatial domain, we adopt $L1$ loss

$$\mathcal{L}_{spa} = \|H_O - GT\|_1 . \tag{3}$$

In frequency domain, we first employ DFT to convert $H_O$ and $GT$ into Fourier space where the amplitude and phase components are calculated. Then, the $L1$-norm of amplitude difference and phase difference between $H_O$ and $GT$ are summed to produce the total frequency loss

$$\mathcal{L}_{fre} = \|\mathcal{A}(H_O) - \mathcal{A}(GT)\|_1 + \|\mathcal{P}(H_O) - \mathcal{P}(GT)\|_1 . \tag{4}$$

Finally, the overall loss function is formulated as follows

$$\mathcal{L} = \mathcal{L}_{spa} + \lambda \mathcal{L}_{fre}, \tag{5}$$

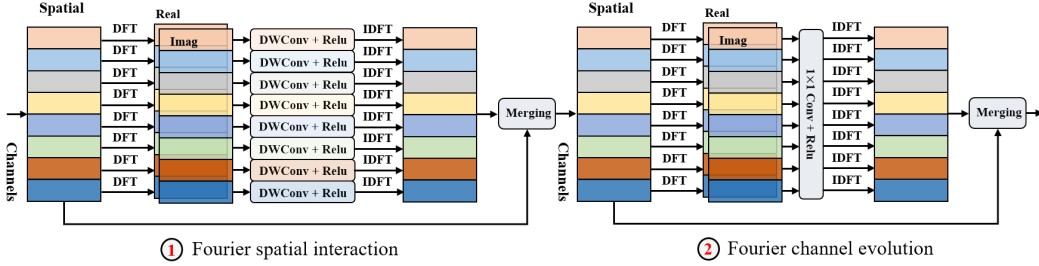

Figure 4: Details of the Fourier Prior Embedded Block. Our block follows the global modeling rule "spatial interaction + channel evolution" but is with new designs: Fourier spatial interaction modeling and Fourier channel evolution.

where $\lambda$ is weight factor and set to 0.1 empirically.

### 3.3 FOURIER PRIOR EMBEDDED BLOCK

As shown in Figure 4, the fundamental building block dubbed as Fourier prior embedded block contains two key elements: (a) Fourier spatial interaction, (b) Fourier channel evolution.

**Fourier spatial interaction.** In terms of the multi-channel feature maps, Fourier transformation is performed independently over each channel. Fourier prior embedded block takes the feature maps as input and then performs Fourier transformation to convert the spatial features into the real and imagery components. Suppose that the features denote as $\mathbf{X} \in \mathrm{R}^{\mathrm{H} \times \mathrm{W} \times \mathrm{B}}$, the corresponding Fourier transformation is expressed as

$$\mathrm{X}_I^{(\mathrm{b})}, \mathrm{X}_R^{(\mathrm{b})} = \mathcal{F}(\mathrm{X}^{(\mathrm{b})}), \tag{6}$$

where $\mathrm{b} = 1, \ldots, \mathrm{B}$, $\mathrm{X}_I^{(\mathrm{b})}$ and $\mathrm{X}_R^{(\mathrm{b})}$ indicate the real and imagery respectively. Then we employ the spatial interaction by a stack of depth-wise convolution with kernel size of $3 \times 3$ and ReLU function. Specifically, $\mathrm{X}_I^{(\mathrm{b})}$ and $\mathrm{X}_R^{(\mathrm{b})}$ share the common depth-wise operator while different channels are independently performed. The spatial interaction can be written as follows:

$$S_I^{(b)} = \sigma \mathbf{DW}^{(\mathbf{b})}(X_I^{(b)}), \tag{7}$$

$$S_R^{(b)} = \sigma \mathbf{DW}^{(\mathbf{b})}(X_R^{(b)}), \tag{8}$$

where $\sigma$ and DW indicate the ReLU function and depth-wise convolution respectively. Next, we apply the inverse DFT to transform the filtered frequency components of $S_I^{(b)}$ and $S_R^{(b)}$ back to spatial domain

$$X_S^b = F^{-1}(S_I^{(b)}, S_R^{(b)}). \tag{9}$$

According to spectral convolution theorem in Fourier theory, processing information in Fourier space is capable of capturing the global frequency representation in frequency domain. Finally, we merge the Fourier spatial interacted feature $X_S$ by concatenating each component $X_S^b$ with the spatial ones processed by the half-instance normalization block, thus generating the output $S_X$.

**Fourier channel evolution.** Followed by the spatial interaction, Fourier channel evolution aims to perform the point-wise channel interaction. Similarly, we first transform the previous step output $S_X$ into the real and imagery components as $C_R$ and $C_I$ and then employ a stack of convolution operator with kernel size of $1 \times 1$ and ReLU function for channel interaction where each position in frequency space is shared. The channel interaction can be written as follows:

$$CX_I = \sigma \mathbf{conv}(cat[C_I^1, \ldots, C_I^B]), \tag{10}$$

$$CX_R = \sigma \mathbf{conv}(cat[C_R^1, \ldots, C_R^B]), \tag{11}$$

where conv indicates the convolution with kernel size of $1 \times 1$. Next, we apply the inverse DFT to transform the filtered frequency components of $CX_I^{(b)}$ and $CX_I^{(b)}$ back to spatial domain as

$$C_S^b = F^{-1}(CX_I^{(b)}, CX_R^{(b)}). \tag{12}$$

Finally, we perform the similar merging process with the first step, thus achieving the global modeling for both spatial and channel dimensions.

# 4 EXPERIMENT

To demonstrate the efficacy of our proposed customized image restoration paradigm, we conduct extensive experiments on multiple computer vision tasks, including image de-raining, image enhancement, image dehazing, and guided image super-resolution. More results can be found in the Appendix.

## 4.1 EXPERIMENTAL SETTINGS

**Low-light image enhancement.** We evaluate our paradigm on two popular benchmarks, including LOL (Chen Wei, 2018) and Huawei (Hai et al., 2021). LOL dataset consists of 500 low-/normal- light image pairs, and we split 485 for training and 15 for testing. Huawei dataset contains 2480 paired images, and we split 2200 for training and 280 for testing. Further, we compared our paradigm with the following 13 state-of-the-art low-light image enhancement methods: SRIE (Fu et al., 2016), RetinexNet (Chen Wei, 2018), MBLLEN (Lv et al., 2018), Enlighten-GAN (Jiang et al., 2021), GLADNet (Wang et al., 2018), Xu et al. (Xu et al., 2020), TBEFN (Lu & Zhang, 2020), KinD (Zhang et al., 2019), Zero-DCE++ (Li et al., 2021), DRBN (Yang et al., 2020), RetinexDIP (Zhao et al., 2021), RUAS (Liu et al., 2021a), KinD++ (Zhang et al., 2021b) and URetinex (Wu et al., 2022).

**Image De-raining.** Following the work (Zamir et al., 2021), our proposed paradigm is evaluated over 13,712 clean-rain image pairs, gathered from multiple synthetic datasets. With this single trained model, we perform evaluation on Rain100H and Rain100L. Further, we report the performance comparison between our designed paradigm and several representative state-of-the-art methods: DerainNet (Yang et al., 2017b), SEMI (Wei et al., 2019), DIDMDN (Zhang & Patel, 2018), UMRL (Yasarla & Patel, 2019), RESCAN (Li et al., 2018b), PReNet (Ren et al., 2019), MSPFN (Jiang et al., 2020), MPRNet (Zamir et al., 2021), HINet (Chen et al., 2021c).

**Image Dehazing.** We evaluate the proposed method on synthetic and real-world datasets. For synthetic scenes, we employ RESIDE (Li et al., 2018a) dataset. The subset Indoor Training Set (ITS) of RESIDE contains a total of 13990 hazy indoor images, generated from 1399 clear images. The subset Synthetic Objective Testing Set (SOTS) of RESIDE consists of 500 indoor hazy images and 500 outdoor ones. In addition, we adopt two real-world datasets: Dense-Haze (Ancuti et al., 2019) and NH-HAZE (Ancuti et al., 2020) to evaluate the generalization. Both of the two datasets consist of 55 paired images. We compare our paradigm with the promising methods: DCP (He et al., 2010) and DehazeNet (Cai et al., 2016), AOD-Net (Li et al., 2017), GridDehazeNet (Liu et al., 2019), FFA-Net (Qin et al., 2020), MSBDN (Dong et al., 2020) and AECR-Net (Wu et al., 2021).

**Guided Image Super-resolution.** Following (Zhou et al., 2022a; Yan et al., 2022), we adopt the pan-sharpening, the representative task of guided image super-resolution for evaluations. The WorldView II, WorldView III, and GaoFen2 in (Zhou et al., 2022a; Yan et al., 2022) are used. To verify the effectiveness of our paradigm, we choose the following representative pansharpening methods for comparison: 1) six state-of-the-art deep-learning based methods, including PNN (Masi et al., 2016), PANNET (Yang et al., 2017a), MSDCNN (Yuan et al., 2018), SRPPNN (Cai & Huang, 2021), GPPNN (Xu et al., 2021b) and INNformer(Zhou et al., 2022a); 2) five promising traditional methods, namely SFIM (Liu., 2000), Brovey (Gillespie et al., 1987), GS (Laben & Brower, 2000), IHS (Haydn et al., 1982), GFPCA (Liao et al., 2017).

Several widely-used image quality assessment (IQA) metrics are employed to evaluate the performance, including the relative dimensionless global error in synthesis (ERGAS) (Alparone et al., 2007), the peak signal-to-noise ratio (PSNR), Structural Similarity Index (SSIM), and the spectral angle mapper (SAM) (J. R. H. Yuhas & Boardman, 1992).

## 4.2 COMPARISON AND ANALYSIS

We perform quantitative performance comparison on the mainstream image restoration tasks in Table 1, Table 2, Table 3, and Table 4, where the best results are highlighted in bold. From the results, it can observed that our proposed paradigm achieves the competitively promising performance with fewer computational burden against the the baselines across all testing datasets on mainstream tasks, suggesting the effectiveness of our designs. For example, for the pan-sharpening, our paradigm

obtains 0.17dB, 0.18dB, and 0.06dB PSNR gains than state-of-art method on the WorldView-II, WorldView-III and GaoFen2 datasets, respectively. In addition, in terms of image enhancement, our paradigm achieves the comparable results with the transformer-based SNRformer with the huge reduce of model parameters and FLOPs. The consistent conclusion can be found in other tasks.

Table 1: **Quantitative comparison of image de-hazing.**

| Method | SOTS | | Dense-Haze | | NH-HAZE | | Param (M) | GFLOPs |
|---|---|---|---|---|---|---|---|---|
| | PSNR | SSIM | PSNR | SSIM | PSNR | SSIM | | |
| DCP | 15.09 | 0.7649 | 10.06 | 0.3856 | 10.57 | 0.5196 | - | - |
| DehazeNet | 20.64 | 0.7995 | 13.84 | 0.4252 | 16.62 | 0.5238 | 0.01M | - |
| AOD-Net | 19.82 | 0.8178 | 13.14 | 0.4144 | 15.40 | 0.5693 | 0.002M | 0.1 |
| GridDehazeNet | 32.16 | 0.9836 | 13.31 | 0.3681 | 13.80 | 0.5370 | 0.96M | 21.5 |
| FFA-Net | 36.39 | 0.9886 | 14.39 | 0.4524 | 19.87 | 0.6915 | 4.68M | 288.1 |
| MSBDN | 33.79 | 0.9840 | 15.37 | 0.4858 | 19.23 | 0.7056 | 31.35M | 41.5 |
| KDDN | 34.72 | 0.9845 | 14.28 | 0.4074 | 17.39 | 0.5897 | 5.99M | - |
| AECR-Net | 37.17 | 0.9901 | 15.80 | 0.4660 | 19.88 | 0.7173 | 2.61M | 43.0 |
| Ours | **37.32** | **0.9901** | **15.95** | **0.4917** | **19.91** | **0.7214** | 1.29M | 20.6 |

Table 2: **Quantitative comparison of image de-raining.**

| Methods | Test100 | | Rain100H | | Rain100L | | Test1200 | | Param (M) | GFLOPs |
|---|---|---|---|---|---|---|---|---|---|---|
| | PSNR↑ | SSIM↑ | PSNR↑ | SSIM↑ | PSNR↑ | SSIM↑ | PSNR↑ | SSIM↑ | | |
| DerainNet | 22.77 | 0.810 | 14.92 | 0.592 | 27.03 | 0.884 | 23.38 | 0.835 | 0.058M | 1.453 |
| SEMI | 22.35 | 0.788 | 16.56 | 0.486 | 25.03 | 0.842 | 26.05 | 0.822 | - | - |
| DIDMDN | 22.56 | 0.818 | 17.35 | 0.524 | 25.23 | 0.741 | 29.65 | 0.901 | 0.373M | 1.686 |
| UMRL | 24.41 | 0.829 | 26.01 | 0.832 | 29.18 | 0.923 | 30.55 | 0.910 | 0.98M | - |
| RESCAN | 25.00 | 0.835 | 26.36 | 0.786 | 29.80 | 0.881 | 30.51 | 0.882 | 1.04M | 20.361 |
| PReNet | 24.81 | 0.851 | 26.77 | 0.858 | 32.44 | 0.950 | 31.36 | 0.911 | 0.17M | 73.021 |
| MSPFN | 27.50 | 0.876 | 28.66 | 0.860 | 32.40 | 0.933 | 32.39 | 0.916 | 13.22M | 604.70 |
| MPRNet | 30.27 | 0.897 | 30.41 | 0.890 | 36.40 | 0.965 | 32.91 | 0.916 | 3.64M | 141.28 |
| HINet | 30.29 | 0.906 | 30.65 | 0.894 | 37.28 | 0.970 | 33.05 | 0.919 | 3.72M | 170.71 |
| Ours | **30.54** | **0.911** | **30.76** | **0.896** | **37.47** | **0.970** | **33.05** | **0.921** | 0.4M | 16.753 |

Table 3: **Quantitative comparison of image enhancement.**

| Method | LOL | | Huawei | | Param (M) | GFLOPs |
|---|---|---|---|---|---|---|
| | PSNR | SSIM | PSNR | SSIM | | |
| SRIE | 12.28 | 0.596 | 13.04 | 0.477 | - | - |
| RobustRetinex | 13.88 | 0.664 | 14.60 | 0.559 | - | - |
| RetinexNet | 16.77 | 0.425 | 16.65 | 0.485 | 0.84M | 148.54 |
| MBLLEN | 17.56 | 0.729 | 16.63 | 0.526 | 0.45M | 21.37 |
| EnGAN | 17.48 | 0.674 | 17.03 | 0.514 | 8.37M | 72.61 |
| GLADNet | 19.72 | 0.680 | 17.76 | 0.521 | 1.13M | 275.32 |
| Xu | 16.78 | 0.766 | 16.12 | 0.586 | 8.62M | 68.45 |
| TBEFN | 17.35 | 0.781 | 16.88 | 0.575 | 0.49M | 24.11 |
| KinD | 20.86 | 0.802 | 16.48 | 0.540 | 8.54M | 36.57 |
| ZeroDCE | 15.29 | 0.518 | 12.46 | 0.407 | 0.08M | 20.24 |
| DRBN | 20.13 | 0.801 | 18.46 | **0.635** | 0.58M | 42.41 |
| RUAS | 16.41 | 0.500 | 13.76 | 0.516 | 0.003M | 0.86 |
| KinD++ | 21.30 | 0.822 | 15.78 | 0.452 | 8.28M | 2970.50 |
| URetinex | 21.32 | **0.835** | 18.79 | 0.607 | 1.23M | 68.37 |
| Ours | **23.57** | 0.832 | **19.17** | 0.621 | 0.08M | 5.03 |

## 4.3 ABLATION STUDIES

To investigate the contribution of the key components, we have conducted comprehensive ablation studies on the WorldView-II satellite dataset of the Pan-sharpening task in terms of the number of network architecture stages and the frequency loss function. More ablated studies can be found in the Appendix.

**Impact of the hierarchical number.** To explore the impact of hierarchical number, i.e., the dom-sampling stages in our U-shape network, we experiment the proposed network with varying num-

Table 4: **Quantitative comparison of guided image super-resolution.**

| Method | worldview II | | | | GaoFen2 | | | | worldview III | | | | Param (M) | GFLOPs |
|---|---|---|---|---|---|---|---|---|---|---|---|---|---|---|
| | PSNR↑ | SSIM↑ | SAM↓ | ERGAS↓ | PSNR↑ | SSIM↑ | SAM↓ | EGAS↓ | PSNR↑ | SSIM↑ | SAM↓ | EGAS↓ | | |
| SFIM | 34.1297 | 0.8975 | 0.0439 | 2.3449 | 36.9060 | 0.8882 | 0.0318 | 1.7398 | 21.8212 | 0.5457 | 0.1208 | 8.9730 | - | - |
| Brovey | 35.8646 | 0.9216 | 0.0403 | 1.8238 | 37.7974 | 0.9026 | 0.0218 | 1.372 | 22.5060 | 0.5466 | 0.1159 | 8.2331 | - | - |
| GS | 35.6376 | 0.9176 | 0.0423 | 1.8774 | 37.2260 | 0.9034 | 0.0309 | 1.6736 | 22.5608 | 0.5470 | 0.1217 | 8.2433 | - | - |
| IHS | 35.2962 | 0.9027 | 0.0461 | 2.0278 | 38.1754 | 0.9100 | 0.0243 | 1.5336 | 22.5579 | 0.5354 | 0.1266 | 8.3616 | - | - |
| GFPCA | 34.5581 | 0.9038 | 0.0488 | 2.1411 | 37.9443 | 0.9204 | 0.0314 | 1.5604 | 22.3344 | 0.4826 | 0.1294 | 8.3964 | - | - |
| PNN | 40.7550 | 0.9624 | 0.0259 | 1.0646 | 43.1208 | 0.9704 | 0.0172 | 0.8528 | 29.9418 | 0.9121 | 0.0824 | 3.3206 | 0.689M | 1.1289 |
| PANNET | 40.8176 | 0.9626 | 0.0257 | 1.0557 | 43.0659 | 0.9685 | 0.0178 | 0.8577 | 29.6840 | 0.9072 | 0.0851 | 3.4263 | 0.688M | 1.1275 |
| MSDCNN | 41.3355 | 0.9664 | 0.0242 | 0.9940 | 45.6874 | 0.9827 | 0.0135 | 0.6389 | 30.3038 | 0.9184 | 0.0782 | 3.1884 | 2.39M | 3.9158 |
| SRPPNN | 41.4538 | 0.9679 | 0.0233 | 0.9899 | 47.1998 | 0.9877 | 0.0106 | 0.5586 | 30.4346 | 0.9202 | 0.0770 | 3.1553 | 17.114M | 21.1059 |
| GPPNN | 41.1622 | 0.9684 | 0.0244 | 1.0315 | 44.2145 | 0.9815 | 0.0137 | 0.7361 | 30.1785 | 0.9175 | 0.0776 | 3.2593 | 1.198M | 1.3967 |
| INNformer | 41.6903 | 0.9704 | 0.0227 | 0.9514 | 47.3528 | 0.9893 | 0.0102 | 0.5479 | 30.5365 | 0.9225 | 0.0747 | 3.0997 | 0.706M | 1.3907 |
| Ours | **41.8325** | **0.9731** | **0.0219** | **0.9506** | **47.5334** | **0.9912** | **0.0102** | **0.5448** | **30.5987** | **0.9241** | **0.0738** | **3.0763** | 0.715M | 1.386 |

Table 5: **Ablation studies for hierarchical number.**

| K | PSNR↑ | SSIM↑ | SAM↓ | ERGAS↓ |
|---|---|---|---|---|
| 1 | 41.1827 | 0.9646 | 0.0255 | 1.0209 |
| 2 | 41.3324 | 0.9655 | 0.0249 | 1.0125 |
| 3 | 41.5331 | 0.9682 | 0.0240 | 0.9839 |
| 4 | 41.6867 | 0.9703 | 0.0235 | 0.9527 |

Table 6: **Ablation studies for frequency loss.**

| Config | PSNR↑ | SSIM↑ | SAM↓ | ERGAS↓ |
|---|---|---|---|---|
| ✗ | 41.7840 | 0.9725 | 0.0221 | 0.9508 |
| ✓ | **41.8325** | **0.9731** | **0.0219** | **0.9506** |

bers. The corresponding quantitative number $K$ comparison from 1 to 4 is reported in Table 5. Observing the results from Table 5, it shows that the model performance can obtain considerable improvements at cost of computation (i.e., large hierarchical number)s. To balance the performance and computational complexity, we set $K = 4$ as default setting for pan-sharpening in this paper.

**Effectiveness of the frequency loss.** The new frequency loss aims to directly emphasize the global frequency information optimization. In Table 6, we remove it to examine its effectiveness. The results in Table 6 demonstrate that removing it severally degrades all metrics dramatically, indicating its significant role in our network.

## 5 LIMITATIONS

First, the more comprehensive experiments on broader computer vision tasks (*e.g.*, image de-noising and image de-blurring) have not been explored. Second, our proposed global modeling paradigm still follows the underlying rule "spatial interaction + channel evolution" of previous transformer-based or MLP-like architectures for general vision tasks. The de facto global modeling rule may be suboptimal for image restoration and it thus needs to be further investigated. In addition, our proposed paradigm has not achieved the best performance. Note that, the objective of our work is orthogonal to previous studies and we thus tailor a simple yet effective and efficient global modelling paradigm for image restoration. This work will spark further research to the realms of the customized global modeling image restoration framework, thus promoting practical application.

## 6 CONCLUSION

In this paper, we first propose a theoretically feasible global modeling paradigm for image restoration. We revisit the existing global modeling paradigm for general vision tasks and find the underlying design rule "spatial interaction + channel evolution". In addition, we revisit the inborn characteristics of Fourier prior for image restoration and find its prevailed decomposed property of image degradation and content component. Based on the above analysis, we customize the core designs: Fourier spatial modeling and Fourier channel evolution. Equipped with above designs, our image restoration paradigm is verified on mainstream image restoration tasks and achieves the competitive performance with fewer computational resources.

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

# Appendix

In this appendix, we provide additional details and results. In Sec. A, we present further discussion on our motivation. In Sec. B, we present the discussion on the reason of our effectiveness. In Sec. C, we show more comparison results between our method and existing methods on multiple image restoration tasks.

## A  MOTIVATION

**Referring to the previous works.** As pointed out in Oppenheim et al. (1979), the motivation comes from a well-known property of the Fourier transformation: the Fourier phase spectrum preserves high-level semantics, while the amplitude spectrum contains low-level features. From Xu et al. (2021a), the amplitude and phase components of Fourier space correspond to the style and semantic information of an image.

**Motivation on image enhancement.** From Xu et al. (2021a), the amplitude and phase components of Fourier space correspond to the style and semantic information of an image. This property can be extended in exposure correction: the amplitude component of an image reflects the lightness representation while the phase component corresponds to structures and is less related to lightness. As shown in Fig. 5, we first swap the amplitude and phase components of different exposures of the same context. More visual clues can refer to Fig. 6 The recombined result of the amplitude of underexposure and the phase of over-exposure has similar lightness appearance with underexposure, while the other behaves conversely. This implies that the swapped amplitude contains most lightness information while the phase component corresponds to the structure representation and is less affected by lightness.

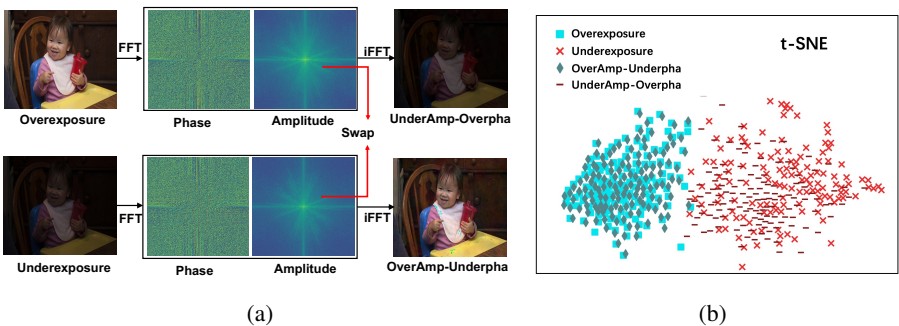

(a)  (b)

Figure 5: **(a) We swap the amplitude and phase components of different exposures of the same context.** The recombined result of the amplitude of underexposure and the phase of over-exposure (UnderAmp-Overpha) has similar lightness appearance with underexposure, while the recombined result of the amplitude of overexposure and the phase of underexposure (OverAmp-Underpha) has similar lightness appearance with overexposure. **(b) The t-SNE for images of over-exposure, underexposure, UnderAmp-Overpha, and OverAmp-Underpha.** The distributions of images in UnderAmp-Overpha and Underexposure are matched, while the distributions of images in OverAmp-Underpha and Overexposure are matched. (b) indicates that the swapped amplitude contains most lightness information.

To further validate our observation, as shown in Fig. 7, we apply the inverse Fast Fourier Transform (iFFT) to the phase and amplitude components to visualize them in spatial domain. The appearance of the phase representation is more similar with the structure representation, and the distribution of the phase component is less affected by lightness. To this end, the phase component is more related to structures that are less affected by lightness in spatial domain.

**Motivation on image de-raining.** Fourier transformation: the Fourier phase spectrum preserves high-level semantics, while the amplitude spectrum contains low-level features Oppenheim et al. (1979). Fig. 8 shows the results of swapping the Fourier amplitude and phase spectrum of rainy/clean images. For the images with or without same content, most rain streaks information

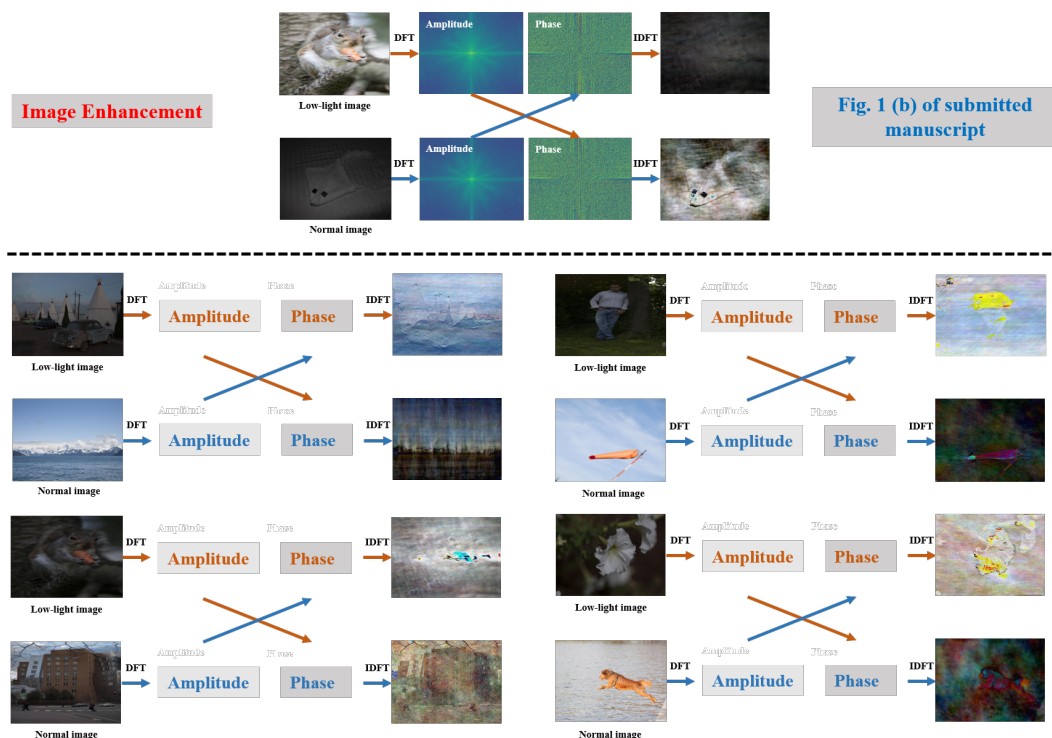

Figure 6: Analysis of discrete Fourier transform (DFT) for low-light image enhancement task. In detail, we swap the amplitude and phase components of the degraded image and a clear version with same or different contents. It can be observed that the degradation effect is transferred with the swapping of amplitude component, indicating that Fourier transform is capable of disentangling image degradation and content component and the degradation mainly lies in the amplitude component. This motivates us to leverage Fourier transform as the image degradation prior embedded into image restoration framework.

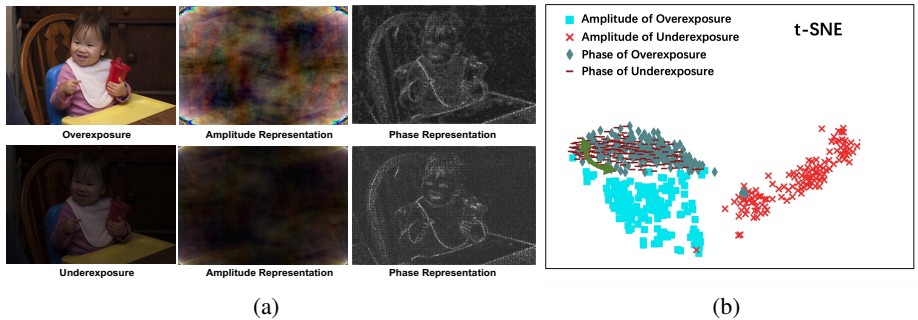

Figure 7: **(a) Visualization of the amplitude and phase components of an image with the same context but different exposures.** We apply the iFFT to the phase and amplitude to compare the phase and amplitude in spatial domain. The amplitude representation significantly differs between different exposures, while the phase representation is very similar across exposures and represents structure representation. **(b) The t-SNE of amplitude and phase of different exposures.** The distributions of phase representations across different exposures are matched, while distributions of amplitude representations across different exposures vary greatly. It means the phase component contains most structure information and is less affected by lightness.

is preserved in the amplitude spectrum of rainy images. This indicates that the phase of rainy im-

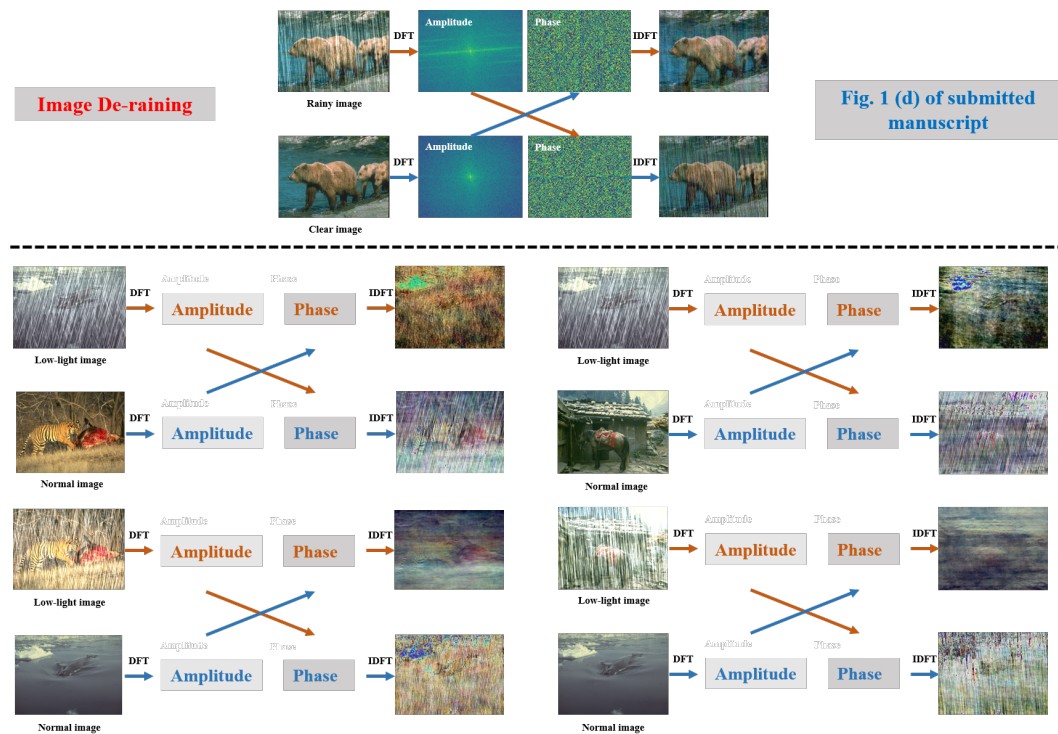

Figure 8: Analysis of discrete Fourier transform (DFT) for image de-raining task. In detail, we swap the amplitude and phase components of the degraded image and the clear version with or without same content. It can be observed that the degradation effect is transferred with the swapping of amplitude component, indicating that Fourier transform is capable of disentangling image degradation and content component and the degradation mainly lies in the amplitude component.

ages keeps the similar background structures as the ground truth. In this way, the Fourier prior is achieved by learning the transformation of the amplitude and phase spectrum separately.

## B    DISCUSSION ON THE REASON OF OUR EFFECTIVENESS

Image restoration is essentially an ill-posed optimization problem. For traditional image restoration algorithms, the common sense is to explore the intrinsic knowledge and image prior to constraint the solution space and thus obtain good solution. Besides, the effectiveness of global modeling for image restoration has been demonstrated in existing works. In our work, our proposed framework incorporates both advantages of global modeling and general image degradation prior that are introduced by Fourier transformation, thus achieving better performance.

Some recent works Dai et al. (2022); Yu et al. (2022) have confirmed that the "spatial interaction + channel evolution" is the core contribution of effectiveness within transformer. Our work stands on the principle with new designs in Fourier space, thus achieving better results.

Image restoration aims to remove the degradation effect and restore clear image. It can be treated as image filtering process. In our work, we conduct extensive analysis in Fourier space and infer that Fourier transform is capable of disentangling image degradation and content component and the degradation mainly lies in the amplitude component. To this end, our method first transforms the spatial representation in Fourier space with amplitude and phase and then employs the convolution to perform the filtering function over the amplitude and phase, thus achieving the clear reconstruction. The Fourier prior is embedded in above procedure and follows the consistent principle of the frequency filtering that is common in digital image processing. Therefore, it further achieves performance gains.

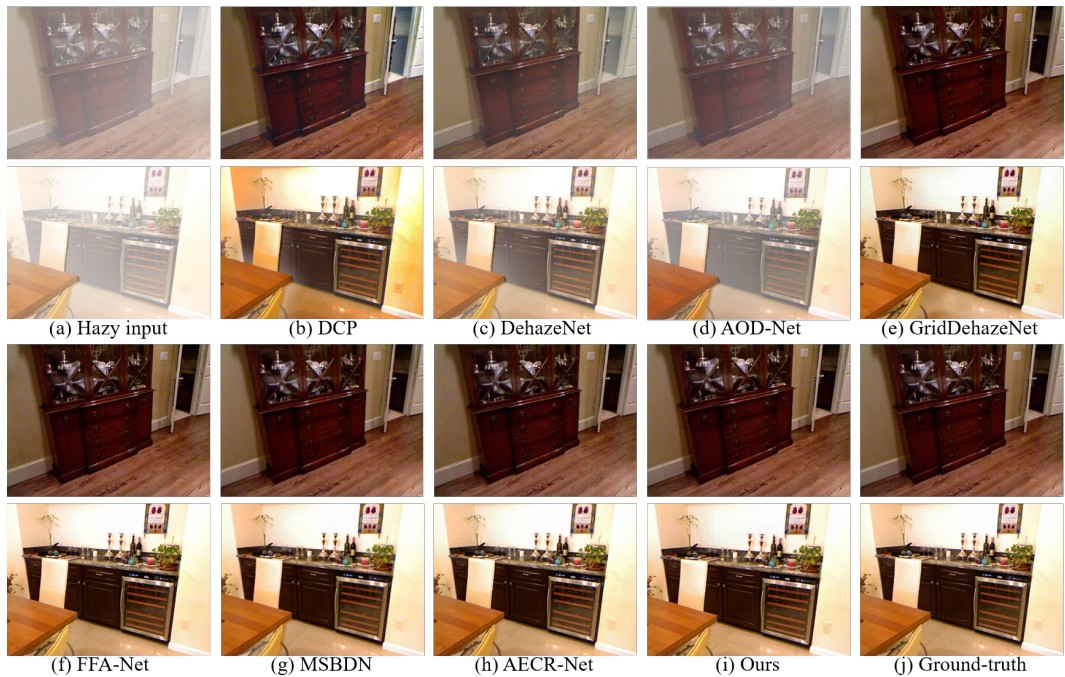

Figure 9: The visual comparison on image de-hazing task.

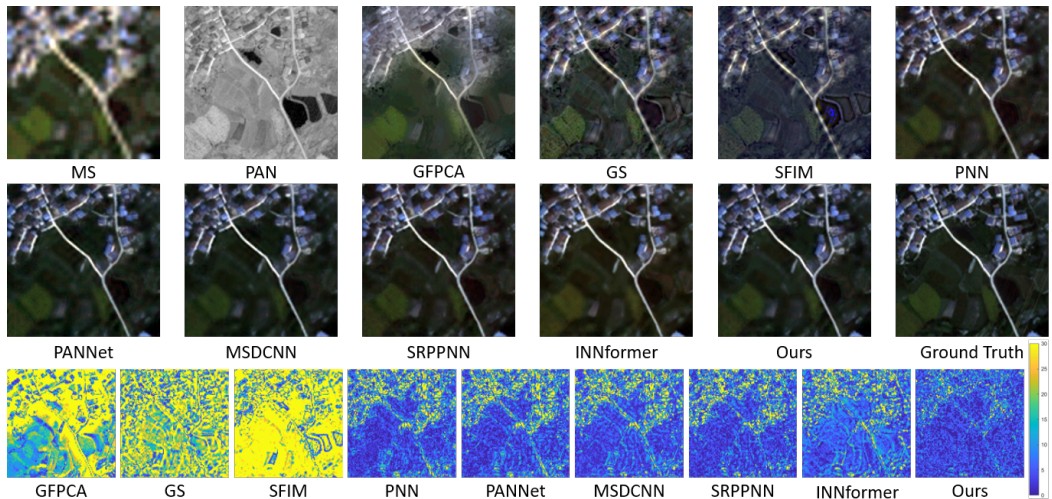

Figure 10: The visual comparison on guided image super-resolution task.

The proposed general image degradation prior is capable of achieving the degradation and content disentanglement, which alleviates the difficulty in network optimization.

## C  MORE COMPARISONS

In this section, we provide more visual comparisons with state-of-the-art methods over the reported tasks. As can be seen in Fig. 9, Fig. 10, Fig. 13, Fig. 14, Fig. 11 and Fig. 12, our proposed method achieves the best performance against other state-of-the-art algorithms.

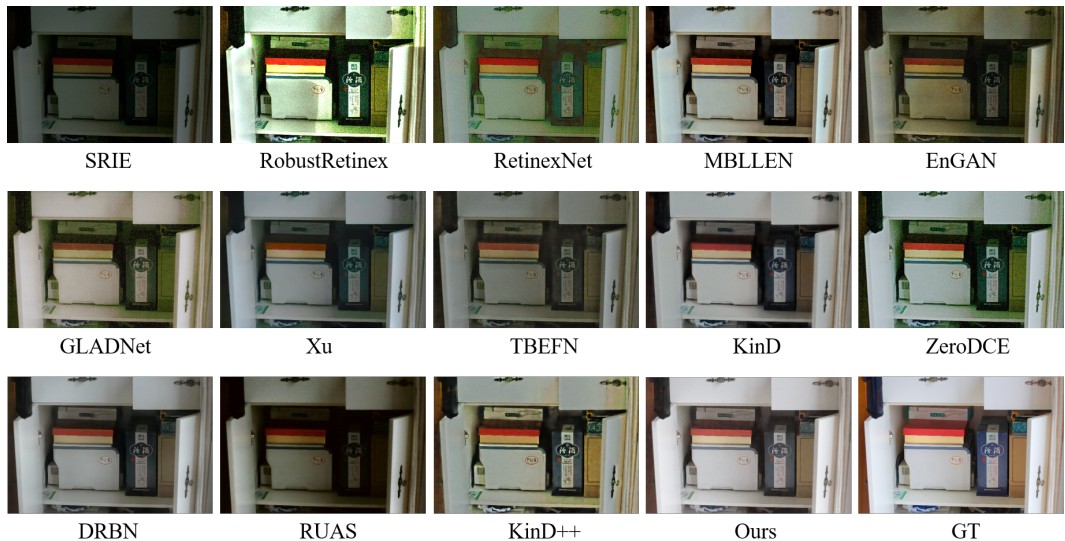

Figure 11: The visual comparison on low-light image enhancement task (LOL dataset).

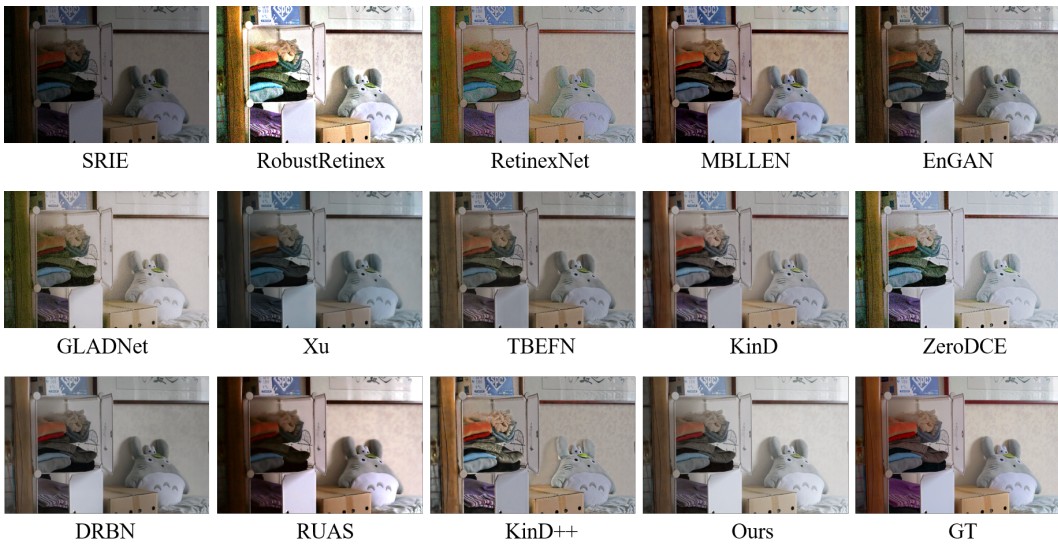

Figure 12: The visual comparison over low-light image enhancement task (LOL dataset).

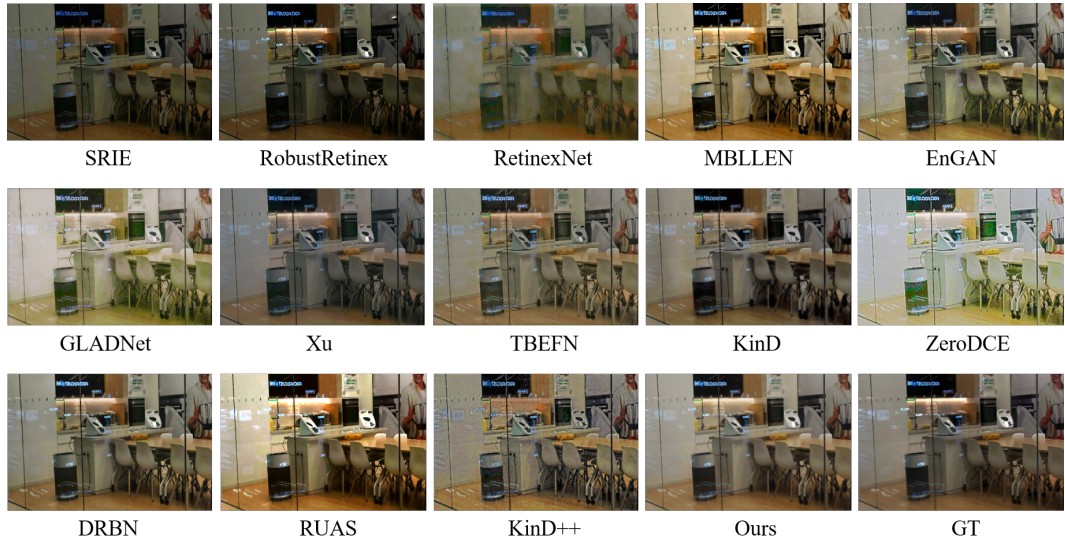

Figure 13: The visual comparison over low-light image enhancement task (Huawei dataset).

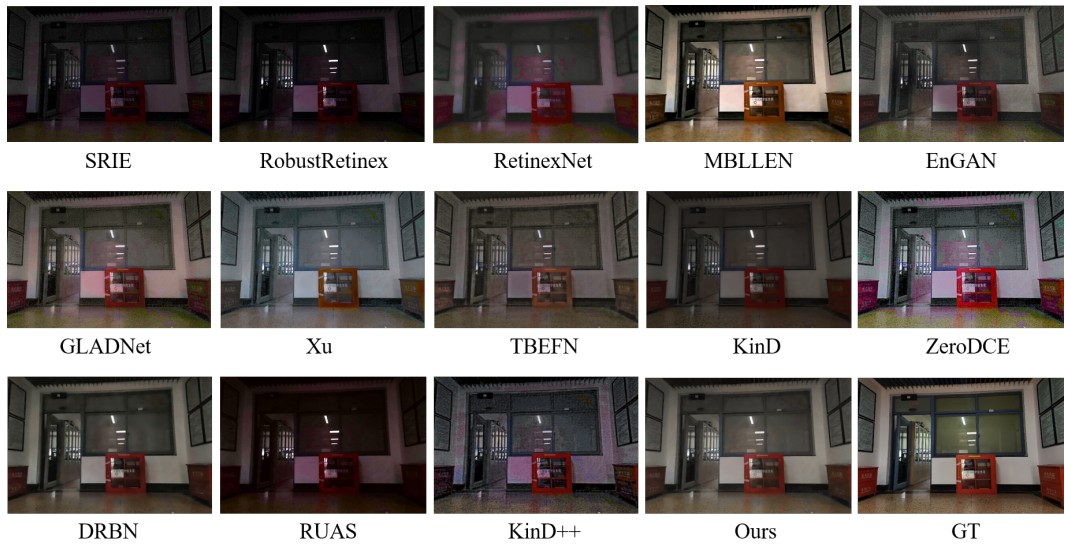

Figure 14: The visual comparison over low-light image enhancement task (Huawei dataset).

