# OpenReview forum: "A simple but effective and efficient global modeling paradigm for image restoration"
_ICLR.cc/2023/Conference — Submitted to ICLR 2023_

### Official Review · Reviewer_FGSi · 2022-10-16

**Confidence:** 4
**Clarity, Quality, Novelty And Reproducibility:** 1. The core idea of this paper is tha…
**Correctness:** 3
**Technical Novelty And Significance:** 4
**Empirical Novelty And Significance:** 3
**Recommendation:** 8

**Strength And Weaknesses:**

This paper proposed a new perspective to deal with different kinds of image degradation problem: that is to say, Fourier transform is capable of disentangling image degradation and content component, acting as the image degradation prior embedded into image restoration framework. Furthermore, the super advantage of the experiments also illustrate the efficiency of the proposed method.

**Summary Of The Paper:**

Different from the global modelling-based image restoration frameworks which provided the remarkable advancement, but at the cost of model parameters and FLOPs while the intrinsic characteristics of specific task are ignored, this paper proposed a simple yet effective global modelling paradigm for image restoration. THe most interesting point of this paper is the proposal of general image restoration prior: Fourier transform is capable of disentangling image degradation and content component, acting as the image degradation prior embedded into image restoration framework. The fruitful experimental results prove the advantage of the proposed paradigm.

**Summary Of The Review:**

In summary, the core idea and the experiments are the advantage. However, the deep analysis of the proposed module can be further enhanced.

---

> ### Author Response · Authors · 2022-11-17
> **Reponse to Reviewer FGSi (part 2/2)**
>
> **2, model parameters.**
>
> In terms of our claims that transformer-based image restoration frameworks suffer from the large model parameters, we have referred to the original papers of the representative works of transformer-based image restoration frameworks. The parameters comparisons are listed as follows, where all the model parameters are directly obtained from the original papers:
>
> |   Model   | #paras |
> |:---------:|:------:|
> |   Uformer | 50.88M |
> | Restormer | 25.31M |
> |   MAXIM   |  22.2M |
> |    Ours   |  0.04M |
>
>
> The large model parameters are mainly in Feed Forward part (FFN) of the transformer structure, where the Multilayer Perceptron (MLP) possesses huge parameters. In addition, these transformer-based methods including the representative Uformer [4], Restormer [5], and MAXIM [6], employ deep network architectures and the deep-in convolution units also have large channel numbers. Therefore, the transformer-based image restoration networks commonly suffer from the huge parameter numbers.
>
> In terms of our proposed paradigm, the main reasons of its effectiveness with fewer computation cost is two folds:
>
> 1) Image restoration is essentially an ill-posed optimization problem. The common sense is to explore the intrinsic knowledge and image prior to constraint the solution space and thus obtain good solution.
>
> 2) Besides, the effectiveness of the transformer-like “spatial interaction + channel evolution” pipeline has been demonstrated in existing works. Our work stands on the principle with new designs in Fourier space, thus achieving better results. In a word, our proposed framework incorporates both advantages of global modeling mechanism and general image degradation prior that are introduced by Fourier transformation, thus achieving better performance. Based on our design, we do not need rough stacking network parameters, but can achieve more rational results through filtering with a small amount of convolution layers.
>
> [4] Uformer: A General U-Shaped Transformer, ICCV 2021.
>
> [5] Restormer: Efficient Transformer for High-Resolution Image Restoration, CVPR 2022.
>
> [6] MAXIM: Multi-Axis MLP for Image Processing, CVPR 2022.

---

> ### Author Response · Authors · 2022-11-17
> **Response to Reviewer FGSi (part 1/2)**
>
> **1, deep analysis for the advantages of the proposed method.**
>
> We have added a section “Discussion on the reason of our effectiveness” to provide the theoretical support in Section B of the Appendix.
>
> Image restoration is essentially an ill-posed optimization problem. For traditional image restoration algorithms, the common sense is to explore the intrinsic knowledge and image prior to constraint the solution space and thus obtain good solution. Besides, the effectiveness of global modeling for image restation has been demonstrated in existing works. In our work, our proposed framework incorporates both advantages of global modeling and general image degradation prior that are introduced by Fourier transformation, thus achieving better performance.
>
> Some recent works [7, 8] have confirmed that the “spatial interaction + channel evolution” is the core contribution of effectiveness within transformer. Our work stands on the principle with new designs in Fourier space, thus achieving better results.
>
> Image restoration aims to remove the degradation effect and restore clear image. It can be treated as image filtering process. In our work, we conduct extensive analysis in Fourier space and infer that Fourier transform is capable of disentangling image degradation and content component and the degradation mainly lies in the amplitude component. To this end, our method first transforms the spatial representation in Fourier space with amplitude and phase and then employs the convolution to perform the filtering function over the amplitude and phase, thus achieving the clear reconstruction. The Fourier prior is embedded in above procedure and follows the consistent principle of the frequency filtering that is common in digital image processing. Therefore, it further achieves performance gains.
>
> The proposed general image degradation prior is capable of achieving the degradation and content disentanglement, which alleviates the difficulty in network optimization.
>
> [7] MetaFormer is Actually What You Need for Vision, CVPR 2022.
>
> [8] Demystify Transformers & Convolutions in Modern Image Deep Networks, arxiv.

---

> ### Author Response · Authors · 2022-11-17
> **To Reviewer FGSi**
>
> We sincerely thank you for reviewing our paper and providing us with valuable feedback. We have addressed your concerns below. The revised parts have been marked in blue in the updated paper and Appendix.

---

### Official Review · Reviewer_8mcG · 2022-10-22

**Confidence:** 5
**Correctness:** 3
**Technical Novelty And Significance:** 4
**Empirical Novelty And Significance:** 3
**Recommendation:** 8

**Clarity, Quality, Novelty And Reproducibility:**

The contributions of this work are significant; the idea is novel and has not seen in previous works; the simple implementation is easy for reproducibility.

**Strength And Weaknesses:**

Strengths:

+ Novel idea: This paper focuses on providing an alternative global modelling-based customized image restoration framework. It contributes the first global modelling paradigm for image restoration in a simple but effective and efficient manner. This work will spark the realms of global modelling-based customized image restoration framework.
+ Interesting analysis and definition: The authors have summarized the de facto global modeling rule of previous studies as the “spatial interaction + channel evolution”. The summary and definition are interesting and impressive. Following the modeling rule and interesting analysis, this work demonstrates that introducing the image prior into global modeling will promote the development of customized global modeling image restoration paradigm.
+ Sufficient experiments and good performance: This paper conducts extensive experiments on many mainstream low-level tasks. The extensive experiments suggest that the proposed paradigm achieves the competitive performance with fewer computational resources.
+ The paper writing is clear and easy to follow.


Weaknesses:

+ The implementation details are confused. For image tasks of image de-hazing/de-raining/enhancement, the proposed paradigm can be directly applied to them as shown in Figure 3. However, in terms of guided image super-resolution with two modalities, the proposed paradigm seems to be modified but the details are not clearly described.
+ Although the proposed paradigm is verified on several image restoration tasks and achieves good quantitative scores under many IQA metrics, it would be better that the visual results are presented.  The reviewer understands the limited space in the main paper. The visual results could be presented in the supplementary material.


**Summary Of The Paper:**

This paper contributes the first global modelling paradigm for image restoration in a simple but effective and efficient manner. The proposed paradigm achieves the competitive performance on several mainstream image restoration tasks with fewer computational resources. It sparks the realms of the global modelling-based customized image restoration framework with promising performance.

**Summary Of The Review:**

This paper proposes to design the global modeling customized image restoration paradigm for the first time and give an alternative solution in a simple but effective and efficient manner. The idea is novel and interesting. I believe that this work will spark the realms of image restoration.  Therefore, I recommend to accept this work.

---

> ### Author Response · Authors · 2022-11-17
> **Response to Reviewer 8mcG**
>
> **1, the implementation details.**
>
> In our work, for single image task, the process pipeline follows the same as the described in Fig. 3 of the manuscript. In terms of guided image super-resolution concerning with multi-modalities, two modalities are concatenated along the channel dimension and then fed into the described pipeline.
>
> **2, visual results.**
>
> Thanks for your suggestion. As suggested, we have provided the visual results in Figures 9-14 of the Appendix.

---

> ### Author Response · Authors · 2022-11-17
> **To Reviewer 8mcG**
>
> We sincerely thank you for reviewing our paper and providing us with valuable feedback. We have addressed your concerns below. The revised parts have been marked in blue in the updated paper and Appendix.

---

### Official Review · Reviewer_DwsU · 2022-10-24

**Confidence:** 5
**Correctness:** 2
**Technical Novelty And Significance:** 2
**Empirical Novelty And Significance:** 2
**Recommendation:** 3

**Clarity, Quality, Novelty And Reproducibility:**

As for the work, the motivation is clear and the authors first propose a theoretically feasible, simple but effective global modeling paradigm
for image restoration. However, the main claims of the paper not at all supported by theory or empirical results.


**Strength And Weaknesses:**

Strength:
1. The authors introduce a newly-designed loss functions to enable the network for better optimization, thus reconstructing the more pleasing results in both spatial and frequency domains.
2. The authors implicitly embed the Fourier-based image prior into the core designs: Fourier spatial modeling and Fourier channel evolution, which achieves the competitive performance on several mainstream image restoration tasks.
Weaknesses:
1. In section Abstract, “the success may be at the cost of model parameters and FLOPs while the intrinsic characteristics of specific task are ignored”, however, compared to previous studies, it is not clear that how to use the intrinsic characteristics of specific task in this work.
2. As for the core designs: Fourier spatial modeling and Fourier channel evolution, but in section 4.3, there is no ablation experiment to verify the specific functions of these two parts, which is critical to this work.
3. Given that the posed paradigm can achieve the competitive performance on several mainstream image restoration tasks with fewer computational resources, but the necessary theoretical support for the good performance is still lacking.


**Summary Of The Paper:**

This paper aims to design a simple but effective and efficient global modeling paradigm for image restoration, which achieves the competitive performance on several mainstream image restoration tasks with fewer computational resources.
The main contributions of this paper are list as follows:
1. The authors contribute the first global modelling paradigm for image restoration in a simple but effective and efficient manner;
2. Following the underlying global modeling rules, this work implicitly embeds the Fourier-based image prior into the core designs: Fourier spatial modeling and Fourier channel evolution.
3. The proposed paradigm achieves the competitive performance on several mainstream image restoration tasks with fewer computational resources.


**Summary Of The Review:**

An ablation study investigates the performance of an AI system by removing certain components to understand the contribution of the component to the overall system. Thus, it is indispensable to do ablation study on the core designs: Fourier spatial modeling and Fourier channel evolution, which is critical to verify the specific functions of these two parts.

---

> ### Author Response · Authors · 2022-11-17
> **Reponse to Reviewer DwsU (part 3/3)**
>
> **3, theoretical support.**
>
> We have added a section “Discussion on the reason of our effectiveness” to provide the theoretical support in Section B of the Appendix.
>
> Image restoration is essentially an ill-posed optimization problem. For traditional image restoration algorithms, the common sense is to explore the intrinsic knowledge and image prior to constraint the solution space and thus obtain good solution. Besides, the effectiveness of global modeling for image restation has been demonstrated in existing works. In our work, our proposed framework incorporates both advantages of global modeling and general image degradation prior that are introduced by Fourier transformation, thus achieving better performance.
>
> Some recent works [7, 8] have confirmed that the “spatial interaction + channel evolution” is the core contribution of effectiveness within transformer. Our work stands on the principle with new designs in Fourier space, thus achieving better results.
>
> Image restoration aims to remove the degradation effect and restore clear image. It can be treated as image filtering process. In our work, we conduct extensive analysis in Fourier space and infer that Fourier transform is capable of disentangling image degradation and content component and the degradation mainly lies in the amplitude component. To this end, our method first transforms the spatial representation in Fourier space with amplitude and phase and then employs the convolution to perform the filtering function over the amplitude and phase, thus achieving the clear reconstruction. The Fourier prior is embedded in above procedure and follows the consistent principle of the frequency filtering that is common in digital image processing. Therefore, it further achieves performance gains.
>
> The proposed general image degradation prior is capable of achieving the degradation and content disentanglement, which alleviates the difficulty in network optimization.
>
> [7] MetaFormer is Actually What You Need for Vision, CVPR 2022.
>
> [8] Demystify Transformers & Convolutions in Modern Image Deep Networks, arxiv.

---

> ### Author Response · Authors · 2022-11-17
> **Reponse to Reviewer DwsU (part 2/3)**
>
> **2, ablation studies.**
>
> 1) As suggested, we have added the corresponding ablation studies as follows. Specifically, we verify the specific function of the designed Fourier block by inserting in into existing methods. In our work, we implement in into three representative baselines PreNet for image de-raining, ZeroDCE for image enhancement, and GPPNN for guided-image super-resolution. Both of them replace the convolution block with our Fourier block.
>
> **Image de-raining**
>
> |        |                   | Rain200L |       | Rain100H |       | Rain100L |       | Test200 |       |
> |:------:|:-------:|:------:|:-------:|:------:|:-------:|:------:|:-------:|:------:|:-------:|
> | method |   config  |   PSNR   |  SSIM |   PSNR   | SSIM  | PSNR     |  SSIM | PSNR    | SSIM  |
> |        |      Original     |   24.81  | 0.851 |   26.77  | 0.858 | 32.44    | 0.950 | 31.36   | 0.911 |
> | PreNet | Fourier prior |   25.56  | 0.868 |   27.35  | 0.861 | 33.85    | 0.963 | 32.45   | 0.922 |
>
> **Guided image super-resolution**
>
> |        |                | WV2 |        |        |        | GaoFen2 |        |        |        | WV3 |        |        |        |
> |:------:|:------:|:-----:|:------:|:------:|:------:|:-----:|:------:|:------:|:------:|:-----:|:------:|:------:|:------:|
> | method |  config|     PSNR     |  SSIM  | SAM    | ERGAS  |   PSNR  | SSIM   | SAM    | ERGAS  | PSNR          | SSIM   | SAM    | ERGAS  |
> |        |    Original    |    41.1622   | 0.9684 | 0.0244 | 1.0315 | 44.2145 | 0.9815 | 0.0137 | 0.7361 | 30.1785       | 0.9175 | 0.0776 | 3.2593 |
> |  GPPNN |  Fourier prior |    41.4513   | 0.9675 | 0.0236 | 1.0001 | 45.5436 | 0.9823 | 0.0135 | 0.6557 | 30.4127       | 0.9201 | 0.0770 | 3.1562 |
>
> **Low-light image enhancement**
> |         |                |  LoL  |       | Huawei |       |
> |:-------:|:--------------:|:-----:|:-----:|:------:|-------|
> |  method |  config |  PSNR |  SSIM |  PSNR  | SSIM  |
> |         |    Original    | 15.29 | 0.518 |  12.46 | 0.407 |
> | ZeroDCE |  Fourier prior | 17.13 | 0.773 |  14.19 | 0.551 |
>
> 2) In addition, we have performed the experiment over image de-raining and guided image super-resolution by removing each of the Fourier spatial interaction and the Fourier channel evolution block to testify its effectiveness.
>
> **Image de-raining**
>
> |                      | Test100 |       | Rain100H |       | Rain100L |       | Test200 |       |
> |:--------------------:|:-------:|:-----:|:--------:|:-----:|:--------:|:-----:|:-------:|:-----:|
> |     configuration    |   PSNR  |  SSIM |   PSNR   |  SSIM |   PSNR   |  SSIM |   PSNR  |  SSIM |
> |         ours         |  30.54  | 0.911 |   30.76  | 0.896 |   37.47  | 0.970 |  33.05  | 0.921 |
> |  w/o Fourier spatial |  30.19  | 0.891 |   30.22  | 0.887 |   36.28  | 0.965 |  32.66  | 0.913 |
> |  w/o Fourier channel |  30.36  | 0.897 |   30.53  | 0.890 |   36.73  | 0.969 |  32.85  | 0.918 |
>
> **Guided image super-resolution**
>
> |                      | WV2 |        |        |        | GaoFen2 |        |        |        |
> |:-----------:|:------:|:------:|:------:|:------:|:-------:|:------:|:------:|:------:|
> |     configuration    |     PSNR     |  SSIM  |   SAM  |  ERGAS |   PSNR  |  SSIM  |   SAM  |  ERGAS |
> |         ours         |    41.8325   | 0.9731 | 0.0219 | 0.9506 | 47.5334 | 0.9912 | 0.0102 | 0.5448 |
> |  w/o Fourier spatial |    41.4513   | 0.9675 | 0.0236 | 1.0001 | 47.1744 | 0.9877 | 0.0106 | 0.6557 |
> |  w/o Fourier channel |    41.5823   | 0.9705 | 0.0227 | 0.9517 | 47.3216 | 0.9892 | 0.0105 | 0.5481 |
>
>
> The results of these ablated models suggest the effectiveness of our Fourier block and the Fourier spatial interaction and Fourier channel evolution.

---

> ### Author Response · Authors · 2022-11-17
> **Reponse to Reviewer DwsU (part 1/3)**
>
> **1, how to use the intrinsic characteristics.**
>
> In our work, we explore general image prior and enable the global modeling framework customized for image restoration. We conduct extensive analysis in Fourier space and infer that Fourier transform is capable of disentangling image degradation and content component and the degradation mainly lies in the amplitude component. This motivates us to leverage Fourier transform as general image degradation prior embedded into image restoration framework. In our work, we use such a general image degradation prior that contains the intrinsic characteristics of specific task for image restoration.
>
> Our method first transforms the spatial representation in Fourier space with amplitude and phase and then employs convolution to perform the filtering function over the amplitude and phase, thus achieving the clear reconstruction. The Fourier prior is embedded in above procedure and follows the consistent principle of the frequency filtering that is common in digital image processing.
>
> The Fourier prior is also exploited in optimization perspective as loss constraint to enable the model generating feasible results.

---

> ### Author Response · Authors · 2022-11-17
> **To Reviewer DwsU**
>
> We sincerely thank you for reviewing our paper and providing us with valuable feedback. We have addressed your concerns below. The revised parts have been marked in blue in the updated paper and Appendix.

---

> ### Author Response · Authors · 2022-12-13
> **Kind Reminder to Reviewer DwsU**
>
> Dear Reviewer,
>
> Thank you again for your feedback. We would like to kindly remind you to confirm whether our changes solved your concerns. If you have any further questions, do not hesitate to reply to our responses. Thank you again for all the efforts that you have made.
>
> If your concerns were solved, could you please consider upgrading the rating? Thank you!
>
> The Authors

---

### Official Review · Reviewer_vJqt · 2022-11-02

**Confidence:** 5
**Clarity, Quality, Novelty And Reproducibility:** See above
**Correctness:** 2
**Technical Novelty And Significance:** 2
**Empirical Novelty And Significance:** 3
**Recommendation:** 3

**Strength And Weaknesses:**

[Pros]
- The paper is clearly written and easy to follow
- Extensive experiments on multiple image restoration tasks looks sound

[Cons]
- The idea of utilizing Fourier transform as a global modeling tool is not new. There are many works that incorporated Fourier transform in the network design but the authors didn't mention (and compare). A closely related competitor is the Fast Fourier Convolution operation proposed in [A], which  explores an efficient ensemble of local and non-local receptive fields in a single unit. This operation is further utilized in image inpainting [B], demonstrating excellent capability to fill large missing areas via global reasoning. Thus, the novelty of the proposed method is quite limited.
- One of the key insight driving this paper is erroneous. The authors claimed "Fourier transform is capable of disentangling image degrada-
tion and content component", this is unfortunately wrong. The examples shown in Figure.1 is misleading. In fact, the Fourier phase component represents the major "appearance" of a image, so swaping the phase component of two images swaps the contents of these two images (You could simply verify this through swaping the phases of two irrelevant images). Consequently, the author's argument "Fourier transform is capable of disentangling image degradation and content component and the degradation mainly lies in the amplitude component" is suspicious and not justified.
- The experimental comparisons against SoTA are not well executed. The authors didn't make "apple to apple" comparisons between the proposed Fourier-based operation and other global modeling operations (e.g., various attention modules). SoTA image restoration methods [C,D] that heavily explore global reasoning are also not compared.

[A] Fast Fourier Convolution, NeurIPS 2020
[B] Resolution-robust Large Mask Inpainting with Fourier Convolutions, WACV 2022
[C] Restormer: Efficient Transformer for High-Resolution Image Restoration, CVPR 2022
[D] MAXIM: Multi-Axis MLP for Image Processing, CVPR 2022

**Summary Of The Paper:**

This paper presents a Fourier-transform-composed building block for neural networks. It follows the popular transformer-like backbone but replaces the non-local attention module with Fourier-transform-based operations. The core idea is to leverage the global property of the Fourier space, i.e., manipulating local areas within Fourier space would affect the global spatial data, as a drop-in replacement for the global feature extraction mechanism typically empowered by computation-intensive attention operations. The experiments on multiple image restoration tasks (enhancement, draining, dehazing, pan-sharpening) collectively demonstrate the superiority of the proposed method against the selected methods in terms of restoration quality and computation efficiency.


**Summary Of The Review:**

Though this paper shows some interesting aspects, the technical and experimental flaws abovementioned prevent it from a positive recommendation.

---

> ### Author Response · Authors · 2022-11-17
> **Reponse to Reviewer vJqt (part 3/3)**
>
> **3, SOTA problem.**
>
> To emphasize, our main focus is not to beat previous frameworks but provide an alternative global modelling-based customized image restoration framework.
>
> Most attention modules are specially designed for high-level vision tasks. The related works cannot be directly used for image restoration tasks. Instead of modifying these works for image restoration that may not make their power full play, we compare our method with the representative state-of-the-art methods in different restoration tasks, some of which already elaborately embed attention mechanism into their networks. In comparison to these methods, the performance of our framework for image restoration is outstanding.
>
> Our work is the first to focus on the customized image restoration framework designs. To emphasize, it is appealing for image restoration algorithms to be deployed on edge devices with limited computation source. Our method has such a potential because of its efficiency.
>
> Different from most of existing computation-intensive attention-based methods, our framework is extremely efficient. We acknowledge that Restormer and MAXIM are good performers for image restoration. Our method is inferior to both methods. However, we wish to emphasize, our framework provides an alternative solution for image restoration when it is efficient with relatively good performance in most image restoration tasks. We list the comparisons between our method and Restormer [4]、Uformer [5],  and MAXIM [6] in terms of parameters as follows.
>
>
> |   Model   | #paras |
> |:---------:|:------:|
> |   Uformer | 50.88M |
> | Restormer | 25.31M |
> |   MAXIM   |  22.2M |
> |    Ours   |  0.04M |
>
> [4] Restormer: Efficient Transformer for High-Resolution Image Restoration, CVPR 2022.
>
> [5] Uformer: A General U-Shaped Transformer, ICCV 2021.
>
> [6] MAXIM: Multi-Axis MLP for Image Processing, CVPR 2022.

---

> ### Author Response · Authors · 2022-11-17
> **Reponse to Reviewer vJqt (part 2/3)**
>
> **2, the key insight.**
>
> In Figure 1 (b), we verify our claim that Fourier transform is capable of disentangling image degradation and content component for image enhancement task. To emphasize, the swapping operating has been performed on two irrelevant images. The visual clues support the claim, i.e., degradation mainly lies in the amplitude component. PLEASE pay attention to Figure 1(b).
>
> To further support our claim, we have provided more visual clues in Figure 5-8 of the Appendix. It is clear that the degradation effect is transferred when swapping the amplitude and phase components of two images with and without same content. The experimental results suggest that Fourier transform is capable of disentangling image degradation and content component and the degradation mainly lies in the amplitude component.
>
> Referring to prior works [1, 2, 3], they have reported the consistent claim with ours.
>
> We further employ the commonly used T-SNE that is tailored for relationship extraction to validate our claim over the above Fourier swap experiments. This results in Figures 5 and 8 of the Appendix also support our claim.
>
> [1] Phase in speech and pictures. ICASSP, 1979.
>
> [2] Exploring Fourier Prior for Single Image Rain Removal. IJCAI, 2022.
>
> [3] A fourier-based framework for domain generalization. CVPR, 2021.

---

> ### Author Response · Authors · 2022-11-17
> **Reponse to Reviewer vJqt (part 1/3)**
>
> **1, the novelty.**
>
> Our main focus is not to beat previous frameworks but wishes to provide an alternative global modelling-based customized image restoration framework. Instead of further exploring the efficacy of Fourier as global modeling in high-level tasks such as image classification, video action classification, human keypoint detection in [A], our work is the first to focus on the customized image restoration framework designs. The work proposed in [A] pays more attention to the global property while our framework further explores the intrinsic prior tailored for image restoration. In a word, different from existing Fourier techniques [B] that emphasize the micro basic operator with global receptive field, our work aims to focus on the macro framework design. In our work, we pay more attention to the customized image restoration global modeling framework. We have cited references [A,B]  that you reported  and discussed the differences in Section 2 of the updated paper.
>
> Our work explores the intrinsic global property of Fourier for image restoration rather than the global modeling. More and more works have focused on the global modeling paradigm. This validates the importance of this issue. Our work explores the issue in image restoration field.
>
> Unlike previous works that only consider the global modeling, our work with efficient structure also meets the requirement of image restoration on edge devices with limited computation sources.

---

> ### Author Response · Authors · 2022-11-17
> **To Reviewer vJqt**
>
> We sincerely thank you for reviewing our paper and providing us with valuable feedback. We have addressed your concerns below. The revised parts have been marked in blue in the updated paper and Appendix.

---

> ### Author Response · Authors · 2022-12-13
> **Kind Reminder to Reviewer vJqt**
>
> Dear Reviewer,
>
> Thank you again for your feedback. We would like to kindly remind you to confirm whether our changes solved your concerns. If you have any further questions, do not hesitate to reply to our responses. Thank you again for all the efforts that you have made.
>
> If your concerns were solved, could you please consider upgrading the rating? Thank you!
>
> The Authors

---

### Author Response · Authors · 2022-11-18
**Kind Reminder**

We appreciate all the reviewers for their valuable time and constructive comments. The comments help us to substantially improve the quality of our paper.

In our revised paper, all the issues that the reviewers raised have been addressed. The changes we made are marked in blue in the revised paper and Appendix and can be summarized as follows:

- We have presented more visual results on different image restoration tasks.
- We have provided more examples and analysis to support our motivations.
- We have emphasized our novelty and discussed the differences from previous works.
- We have compared the model size of our method with the transformer-based and MLP-based methods.
- We have provided more details and settings of our method and experiments.
- We have added more ablation studies to verify the effectiveness of our designs.
- Code will be released. We will also provide the ablated models and the models pre-trained on different datasets.

Since it is close to the end of the discussion period, we would like to kindly remind the reviewers to confirm whether our changes solved your concerns. If you have any further questions, do not hesitate to reply to our responses. Thank you again for all the efforts that you have made.


Best wishes,

Authors

---

### Decision · Program_Chairs · 2023-01-20

**Decision:**

Reject

**Justification For Why Not Higher Score:**

The rebuttal makes it clear the authors do not intend to compare to SOTA works, despite other 30M parameter models being in the current comparison.

**Justification For Why Not Lower Score:**

n/a

**Metareview: Summary, Strengths And Weaknesses:**

The paper proposes a model which makes use of Fourier transform modules in image restoration.  The potential advantage of this choice is that a model with relatively few parameters (say 40K) can demonstrate high quality results, beating some previous models which are considerably larger.  This is a potentially valuable contribution, but as I argue now, the paper needs a much more thoughtful and thorough re-presentation in order to be published at the level of this conference.

As pointed out by reviewer vjqt, the results are surpassed by other existing methods, and the authors make clear in the rebuttal and revised paper that (a) these systems perform better - "our method is inferior to such methods", and (b) they do not intend to compare to those methods in the final paper, nor even make the clear statement that they perform better.  The rebuttal suggests that this is because those models have 20M+ parameters, but the paper already includes models up to 30M parameters, so that selection criterion is not consistent with what has already been written.

Several times in the rebuttal or in the updated paper the authors make claims such as " our proposed framework is the first to explore the customized image restoration global modeling paradigm." or  "the objective of our work is orthogonal to previous studies and we thus tailor a simple yet effective global modelling paradigm for image restoration." or  "our main focus is not to beat previous systems".  The authors appear to be arguing that their intentions are more important than the artefacts they have created or are describing.  This is simply not the case.  The proposed system has potential utility: it may be faster or more efficient than other models, at a cost of reduced accuracy.  In order for readers to evaluate the utility of the work, it needs to be made clear what the increase in efficiency is, *and* the reduction in accuracy.  Similarly, with a claim such as  "all above architectures are not fully to explore priors that are specific for image restoration tasks, which is important to lift performance." This may be true, but it needs to be *shown* that is is important to "lift performance" by running the appropriate experiments, and by comparing against the other systems.  The new ablations go some way toward answering this question, showing perhaps 0.5-1.5 PSNR improvements, significant, but only on "middle-performing" models.   To be clear: I am not suggesting that to be accepted this paper needs to beat the SOTA systems for accuracy.  It does not.  However it must *show* the reduction in order to show what is lost by the gain in efficiency.

In conclusion, the review and rebuttal process has clearly led to improvements in the paper, however the revision as is does not adequately take into account the reviewers' criticisms.  I would strongly encourage the authors to recast the paper to show clearly how their proposal can lead to an efficient system which is not state of the art in terms of accuracy.  A pareto front on a (speed vs accuracy) graph might serve that purpose.

The authors are concerned that reviewer vjqt has misunderstood their figure 1b.  This does not reduce the accuracy of the reviewer's discussions on experimental comparisons.